**Cyanobacterial carbon concentrating mechanisms facilitate sustained $CO_2$ depletion in**
**eutrophic lakes**
Ana M. Morales-Williams[1,2,3], Alan D. Wanamaker[4], Jr., and John A. Downing[1,5]
[1]Department of Ecology, Evolution, and Organismal Biology, Iowa State University, 251 Bessey
Hall, Ames, IA, 50011, USA
[2]Department of Ecology, Evolution, and Behavior, University of Minnesota-Twin Cities, 1475
Gortner Ave., Saint Paul, MN, 55108, USA
[3]Rubenstein School of Environment and Natural Resources, University of Vermont, 81 Carrigan
Drive, Burlington, VT, 05405
[4]Department of Geological and Atmospheric Science, Iowa State University, 12 Science 1,
Ames, IA, 50011, USA
[5]Minnesota Sea Grant, University of Minnesota-Duluth, 141 Chester Park, 31 West College St.,
Duluth, MN, 55812, USA
**Correspondence:** Ana M. Morales-Williams, ana.morales@uvm.edu
**Abstract**

23        Phytoplankton blooms are increasing in frequency, intensity, and duration in aquatic

ecosystems worldwide. In many eutrophic lakes, these high levels of primary productivity
correspond to periods of $CO_2$ depletion in surface waters.  Cyanobacteria and other groups of
phytoplankton have the ability to actively transport bicarbonate ($HCO_3^-$) across their cell
membrane when $CO_2$ concentrations are limiting, possibly giving them a competitive advantage
over algae not using carbon concentrating mechanisms (CCMs). To investigate whether CCMs
can maintain phytoplankton bloom biomass under $CO_2$ depletion, we measured $\delta^{13}C$ signatures
of dissolved inorganic carbon ($\delta^{13}C_{DIC}$) and phytoplankton particulate organic carbon ($\delta^{13}C_{phyto}$)
in sixteen mesotrophic to hypereutrophic lakes during the ice-free season of 2012. We used mass
balance relationships to determine the dominant inorganic carbon species used by phytoplankton
under $CO_2$ stress. We found a significant positive relationship between phytoplankton biomass
and phytoplankton $\delta^{13}C$ signatures, as well as a significant non-linear negative relationship
between water column $\rho CO_2$ and isotopic composition of phytoplankton, indicating a shift from
diffusive uptake to active uptake by phytoplankton of $CO_2$ or $HCO_3^-$ during blooms. Calculated
photosynthetic fractionation factors indicated that this shift occurs specifically when surface
water $CO_2$ drops below atmospheric equilibrium. Our results indicate active $HCO_3^-$ uptake via
CCMs may be an important mechanism maintaining phytoplankton blooms when $CO_2$ is
depleted.  Further increases in anthropogenic pressure, eutrophication, and cyanobacteria blooms
are therefore expected to contribute to increased bicarbonate uptake to sustain primary
production.
**Key words:** Eutrophication, carbon cycling, cyanobacteria, CCM, stable isotopes

## 1. Introduction

Cyanobacteria blooms resulting from anthropogenic eutrophication are among the greatest current threats to inland water ecosystems, altering carbon cycling and ecosystem function, impairing water quality, and endangering human health (Brooks et al., 2016; Paerl et al., 2011; Visser et al., 2016). Forecasting models and macrosystem-scale analyses suggest the occurrence of blooms is driven by the interactive effects of land use, nutrient inputs (nitrogen and phosphorus), climate, weather, and in-lake processes (Anneville et al., 2015; Michalak et al., 2013; Persaud et al., 2015; Rigosi et al., 2014). Mechanisms determining variability in timing and duration of these events in  lakes, however, remain poorly understood (Brooks et al., 2016), and it is unclear what the large-scale feedbacks of sustained primary production are on lake carbon cycling by phytoplankton. While temperate lakes have generally been considered net sources of $CO_2$ to the atmosphere (Tranvik et al., 2009), eutrophic systems can maintain both high levels of primary production and negligible concentrations of $CO_2$ in surface water (Balmer and Downing, 2011; Gu et al., 2010; Laas et al., 2012), possibly increasing the flow of dissolved inorganic C to organic C. Identifying drivers of the temporal variability of bloom formation and maintenance will contribute to a better understanding of carbon dynamics in lakes with high productivity.

Cyanobacteria have developed a suite of diverse strategies for obtaining and fixing carbon and nutrients at growth-limiting concentrations. In addition to fixing atmospheric nitrogen, they are able to maintain metabolic processes under severe $CO_2$ depletion by use of a carbon concentrating mechanism (CCM; Badger and Price 2003; Raven et al. 2008). The cyanobacterial CCM is not only the accumulation of inorganic carbon, but collectively active transport across the cell membrane, partitioning of Rubisco into carboxysomes, and elevation of $CO_2$ around

enzyme complexes (Price et al., 2008b). When water column pH exceeds 8.5, $CO_2$ is negligible
and $HCO_3^-$ is the dominant carbon species. $HCO_3^-$ cannot passively diffuse across phytoplankton
cell membranes, and therefore requires an active transport system. CCMs are present in many
groups of aquatic photoautotrophs including green algae (Spalding, 2008) and diatoms
(Hopkinson et al., 2016), as well as some higher plants. These mechanisms are thought to have
evolved independently in eukaryotic algae and the cyanobacteria, corresponding to a large
decrease in atmospheric $CO_2$ and doubling of $O_2$ approximately 400 million years ago (Badger
and Price, 2003; Raven et al., 2008). There are, however, many similarities between eukaryotic
and cyanobacteria CCMs which are not fully resolved, so it is unclear whether or not
cyanobacteria CCMs represent a more efficient, competitive advantage over other phytoplankton
taxa (Moroney and Ynalvez, 2007).
The cyanobacterial CCM mechanism facilitates active transport of $HCO_3^-$ across the
plasma membrane, where it is accumulated in the cytosol, transferred to Rubisco-containing
carboxysomes, and converted to $CO_2$ via carbonic anhydrases (Raven et al., 2008). Carboxysome
structures, unique to cyanobacteria CCMs, are thought to decrease $CO_2$ leakage rates via low
permeability for uncharged species (i.e., $CO_2$) across the carboxysome protein shell (Kaplan and
Reinhold, 1999; Price et al., 2008a). In an optimal CCM, diffusion of $HCO_3^-$ across the
carboxysome shell is fast, and leakage of converted $CO_2$ is slow (Mangan and Brenner, 2014).
This results in reduced isotopic discrimination and an intracellular composition approaching that
of source material (Fielding et al., 1998).
In freshwaters, cyanobacteria use form 1B Rubisco, which facilitates acclimation to
inorganic carbon depletion via high cellular affinity for $CO_2$ and $HCO_3^-$ (Raven and Beardall,
2016; Raven et al., 2008; Shih et al., 2015). While this process is energetically costly, it is
essential to both increase photosynthetic efficiency and local bioavailability of inorganic carbon
when $CO_2$ is depleted. In addition to inorganic carbon availability, cyanobacterial CCMs are
triggered by photosynthetically active radiation (PAR) and nitrogen availability. Because CCMs
are energetically costly (Raven and Beardall, 2016), decreased PAR lowers cellular affinity for
inorganic carbon (Giordano et al., 2005). Affinity increases with depletion of nitrate and iron, but
decreases with depletion of $NH_4^+$, and does not have a consistent response to phosphorus
limitation (Raven et al., 2008).  CCM activation under carbon and nutrient stress thus may confer
a competitive advantage to cyanobacteria via efficient carbon fixation when $CO_2$ is low (Badger
and Price, 2003; Price et al., 2008b).

Shifts to alternative carbon assimilation strategies result in measureable changes in

isotopic fractionation. Stable isotopic signatures of phytoplankton are dependent both on the
isotopic composition of their DIC source and the physiological mechanism used to acquire it.
When phytoplankton use passive diffusion to take up ambient $CO_2$, photosynthetic fractionation
resembles that of C3 terrestrial plants (Yoshioka, 1997), resulting in typical mean $\delta^{13}C$ signatures
between -27‰ to -30‰ (Bade et al., 2004; Erez et al., 1998; O'Leary, 1988). In cyanobacteria
and other phytoplankton, carbon fixation can be equally limited by carboxylation and active
inorganic carbon transport into the cell. Cyanobacteria and eukaryotic algae that are actively
concentrating inorganic carbon via $HCO_3^-$ uptake can have elevated $\delta^{13}C$ values as high as -8 to -
11‰ (Sharkey and Berry, 1985; Vuorio et al., 2006). This is largely attributable to the isotopic
signature of source material (Kaplan and Reinhold, 1999), as well as decreased carbon efflux
when CCMs are active, resulting in reduced photosynthetic fractionation (-1‰ to -3‰; Sharkey
and Berry 1985; Erez et al. 1998).  Further, isotopic fractionation associated with active $HCO_3^-$
uptake is negligible (Sharkey and Berry, 1985; Yoshioka, 1997). In other words, discrimination
due to passive diffusion is reduced or negligible when active $HCO_3^-$ uptake is occurring
(Giordano et al., 2005). Thus, if CCMs are activated during cyanobacteria blooms in eutrophic
lakes, we would expect the $\delta^{13}C$ signature of the phytoplankton to increase as ambient $CO_2$ is
depleted, and photosynthetic fractionation factors to decrease as the community becomes
dominated by phytoplankton using CCM.
The purpose of this study was to evaluate the importance of CCMs in maintaining high
phytoplankton biomass during $CO_2$ depletion in eutrophic and hypereutrophic lakes. We
hypothesized that photosynthetic fractionation would be tightly coupled with inorganic carbon
limitation, resulting in decreased fractionation with shifts from atmospheric $CO_2$ to mineral
$HCO_3^-$ in the water column. We further hypothesized that phytoplankton isotopic composition
and photosynthetic fractionation would correspond to $CO_2$ depletion in the water column,
reflecting CCM activation during blooms that are intense enough to lower water column $CO_2$.
**2. Methods**
16 lakes were chosen based on Iowa State Limnology Laboratory long-term survey data
(total phosphorus and phytoplankton community composition, 2000-2010, data publically
available via the Iowa Department of Natural Resources Lake Information System:
http://limnology.eeob.iastate.edu/lakereport/) along an orthogonal gradient of watershed
permeability (Fraterrigo and Downing, 2008) and interannual variability in cyanobacteria
dominance. Long term survey data were used only for site selection. Duplicate stable isotope
samples for particulate organic and dissolved inorganic analyses were collected once following
ice off in 2012, weekly May-July, bi-weekly in August, and monthly September-November
($n$=196). Standard physical, chemical, and biological parameters were measured at each
sampling event using US-EPA certified methods, including total nitrogen (TN), total phosphorus
(TP), chlorophyll a (Chl a), alkalinity and pH. Samples for phytoplankton community
characterization were collected three times during the summer in each lake using a vertical
column sampler from the upper mixed layer. Aqueous carbon dioxide concentration was
measured at 1 m using a Vaisala GMT2220 probe modified for water measurements (Johnson et
al., 2009). Partial pressure of carbon dioxide ($pCO_2$) was determined using temperature, depth,
and pressure corrections described in Johnson et al.( 2009). Specifically, because pressure and
temperature respectively increase and decrease sensor output relative to their calibration,
measurements were reduced by 0.15% per unit increase hPa relative to calibration (1013 hPa),
and increased 0.15% per unit hPa decrease. An additional correction for depth was added to the
barometric pressure correction, because pressure is increased 9.81 hPa per 10 cm depth.
Measurements were taken at 1 m, equivalent to a 98.1 hPa increase. Similarly, measurements
were increased by 0.3% per degree Celsius increase in water temperature above instrument
calibration (25°C).

All water chemistry was performed in the Iowa State Limnology Laboratory using United

States Environmental Protection Agency (US EPA) certified methods. Total nitrogen was
determined using the second derivative method described in Crumpton et al. (1989). Total
phosphorus was determined colorimetrically using the molybdate blue method (APHA, 2012).
Samples for Chl *a* analysis were filtered onto GF/C filters which were frozen then extracted and
sonicated in cold acetone under red light. Samples were then analyzed fluorometrically (Arar and
Collins, 1997; Jeffrey et al., 1997). Alkalinity was determined by acid titration and reported as
mg $CaCO_3$ $L^{-1}$ (APHA, 2012). Field measurements of temperature, DO, pH, and conductivity
were taken with a YSI multi-parameter probe.

Phytoplankton community and biomass samples reported here were processed and

analyzed in the Iowa State Limnology Laboratory. These data can also be accessed via the Iowa
Department of Natural Resources Lake Information System. Samples were counted to 150
natural units of the most abundant genera, and biovolume determined following Hillebrand et al.
(1999). Biomass was determined from biovolume assuming cell density of 1.1 g cm$^{-3}$ (Filstrup et
al., 2014; Holmes et al., 1969).

Samples collected for isotopic analysis of dissolved inorganic carbon ($\delta^{13}C_{DIC}$) were

filtered to 0.2 µm in the field using a syringe filter and cartridge containing a combusted GF/F
prefilter (Whatman) and 0.2 µm polycarbonate membrane filter (Millipore). Samples were then
injected into helium gas-flushed septa-capped vials with $H_3PO_4$ to cease biological activity and
to sparge $CO_2$ (Beirne et al., 2012; Raymond and Bauer, 2001). $\delta^{13}C_{DIC}$ samples were measured
via a Finnigan MAT Delta Plus XL mass spectrometer in continuous flow mode connected to a
Gas Bench with a CombiPAL autosampler. Reference standards (NBS-19, NBS-18, and LSVEC)
were used for isotopic corrections, and to assign the data to the appropriate isotopic scale
(Vienna Pee Dee Belemnite, VPDB, for carbonates). Average analytical uncertainty (analytical
uncertainty and average correction factor) was ±0.06 ‰ (1 sigma, VPDB). Samples were
analyzed by standard isotope ratio mass spectrometry methods (IRMS), and reported relative to
VPDB in ‰ (Equation 1).
$\delta^{13}C_{Sample} = [(^{13}C/^{12}C)_{sample}/(^{13}C/^{12}C)_{VPDB} - 1] \times 1000$        Eq. 1

To determine the isotopic composition of phytoplankton organic carbon ($\delta^{13}C_{phyto}$),

samples were filtered onto pre-combusted GF/C filters. Zooplankton and detritus were removed
manually from filtered samples using a dissecting microscope. Samples were gently fumed in a
desiccator for 24 h with 1N HCl to remove inorganic carbon, dried in a low temperature oven,
then pulverized using a mortar and pestle and analyzed with standard methods (above IRMS
connected to a Costech Elemental Analyzer).  Calcification is common in marine phytoplankton,
but not in eutrophic freshwater lakes and was not observed in our samples. For organic isotope
samples, three reference standards (Caffeine [IAEA-600], Cellulose [IAEA-CH-3], and
Acetanilide [laboratory standard]) were used for isotopic corrections, and to assign the data to
the appropriate isotopic scale (VPDB for carbonates). The average combined uncertainty for
$\delta^{13}C$ was $\pm$ 0.17‰ (1 sigma, VPDB). For all isotopic measurements, at least one reference
standard was used for every six samples.

Photosynthetic fractionation factors of biomass relative to ambient $CO_2$ ($\varepsilon_p$) were

calculated using published temperature dependent fractionation factors between carbon species
following methods described in Trimborn et al. 2009 (Mook, 1986; Trimborn et al., 2009),
reflecting cumulative fractionation occurring during phytoplankton growth.  Inorganic carbon
fractions and total DIC concentration were calculated using discrete $CO_2$, alkalinity, and pH
measurements:

$\delta^{13}C_{HCO_{3-}} = \dfrac{\delta^{13}C_{DIC}[DIC]-(\varepsilon_a[CO_2]+\varepsilon_b[CO_3^{2-}])}{(1+\varepsilon_a*10^{-3})[CO_2]+[HCO_{3-}]+(1+\varepsilon_b*10^{-3})[CO_3^{2-}]}$         Eq. 2
$\delta^{13}C_{CO2}= \delta^{13}C_{HCO3-} (1 + \varepsilon_a \times 10^{-3}) + \varepsilon_a$         Eq. 3
$\varepsilon_p = (\delta^{13}C_{CO2} - \delta^{13}C_{phyto}) / (1 + (\delta^{13}C_{phyto} / 1000))$         Eq. 4
where $\varepsilon_a$ and $\varepsilon_b$ are temperature dependent fractionation factors between $CO_2$ and $HCO_3^-$, and
$HCO_3^-$ and $CO_2^{3-}$, respectively (Trimborn et al. 2009, as referenced therein).

To test the hypothesized relationships between phytoplankton isotopic composition,

photosynthetic fractionation, and ambient $pCO_2$ (n=196), we used a nonlinear dynamic
regression and ran 199 model iterations (SigmaPlot 12, Systat Software) resulting in 100%
model convergence. We used linear regression to test the relationship between photosynthetic
fractionation ($\varepsilon_p$) and the isotopic composition of the DIC pool. The relationship between
phytoplankton biomass as chlorophyll *a* (Chl *a*) and phytoplankton isotopic composition using a
Pearson correlation. Prior to analyses, data were tested for normality using a Shapiro Wilk test.
**3. Results**

Phytoplankton biomass during productive summer months (May-August) ranged from 4.3

mg L$^{-1}$ in Springbrook Lake in August to 4120.35 mg L$^{-1}$ in Lake Orient in June. Phytoplankton
communities were consistently dominated by cyanobacteria with the exceptions of East Lake
Osceola in June and August and Springbrook Lake in August, which were both dominated by
diatoms (Figures 1 and 2). Maximum cyanobacteria biomass was measured in Lake Orient in
June (4119.34 mg L$^{-1}$) and the minimum occurred in Silver Lake-D in August (3.70 mg L$^{-1}$).

Phytoplankton $\delta^{13}C$ signatures in this study ranged from -29.86 ‰ to -13.48 ‰ with an

average -25.26 ± 2.8 ‰. The highest values were measured when algal biomass peaked (i.e.,
during summer months, Table 2). Overall, pH increased slightly and $CO_2$ decreased during
blooms relative to non-bloom conditions (Tables 1 and 2).  All lakes except Arrowhead and
George Wyth experienced cyanobacteria blooms. Phytoplankton $\delta^{13}C$ and phytoplankton
biomass inferred from Chl a concentration were positively correlated (Pearson correlation, µg
Chl *a* L$^{-1}$,  $R = 0.60$, $P < 0.001$, Figure 3), suggesting a shift from diffusive to active uptake of
inorganic carbon during blooms. Over the course of this study, bloom conditions, defined as > 40
µg Chl *a* L$^{-1}$ (Table 1; Bachmann et al. 2003), were observed in 46% of our observations with
varying degrees of intensity. TN and TP measured across the study were on average in the
eutrophic to hypereutrophic range (Table 1).
To evaluate the predicted shift in algal carbon assimilation strategies below atmospheric
equilibrium, we used a nonlinear dynamic model to analyze the relationships between ambient
$pCO_2$ and $\delta^{13}C_{phyto}$ across lakes and sampling events. We found that while no relationship existed
between these variables above atmospheric equilibrium, there was a rapid, significant increase in
$\delta^{13}C_{phyto}$ (Figure 4, top; $R^2$=0.58, $P$<0.001) and decrease in fractionation (Figure 4, bottom;
$R^2$=0.66, $P$<0.001) as $CO_2$ was depleted below atmospheric equilibrium (393 ppm, NOAA Earth
System Research Laboratory, http://www.esrl.noaa.gov/). We found a significant, positive, linear
relationship between the stable isotopic composition of the DIC pool and photosynthetic
fractionation ($\varepsilon_p$, $R^2$=0.72, P<0.001, Figure 5). Relationships between $pCO_2$ and $\delta^{13}C_{phyto}$ for
individual lakes can be found in supplemental information (Figures S1 and S2).
**4. Discussion**
Our results indicate that alternative carbon assimilation strategies may be an important
mechanism sustaining cyanobacteria blooms in anthropogenically eutrophic and hypereutrophic
lakes. Here we demonstrate that the relationship between $pCO_2$ and photosynthetic fractionation
exists only when $pCO_2$ drops below atmospheric equilibrium during blooms. We found a similar
clear breakpoint below atmospheric equilibrium between $pCO_2$ and phytoplankton isotopic
composition, together suggesting that CCM mechanisms are switched on in phytoplankton
communities when ambient water column $CO_2$ is depleted below atmospheric levels.
While previous models found no predictive relationship between ambient $pCO_2$ and
photosynthetic fractionation (Bade et al., 2006), other proxy-based studies have shown long term
relationships between $pCO_2$ and the isotopic composition of phytoplankton (Smyntek et al.,
2012).The range of values measured in our study for both $\delta^{13}C_{phyto}$ and $\varepsilon_p$ is consistent with
previous laboratory and marine field studies demonstrating shifts from diffusive to active
inorganic carbon assimilation via CCM activation (Boller et al., 2011; Cassar, 2004; Erez et al.,
1998; Trimborn et al., 2009). Calculated photosynthetic fractionation was lowest during blooms,
consistent with phytoplankton CCM utilization. While previous freshwater studies have
demonstrated similar variability in phytoplankton isotopic composition (Vuorio et al., 2006),
ours is the first to demonstrate the co-occurrence of decreased fractionation with $CO_2$ depletion
during blooms in eutrophic and hypereutrophic lakes. The cellular mechanisms contributing to
the decrease in fractionation likely provide a competitive advantage to bloom-forming taxa when
high productivity depletes ambient $CO_2$.

In eutrophic lakes, both phytoplankton isotopic composition and fractionation appear to be

strongly related to $pCO_2$ availability below a critical equilibrium point. In less productive
northern temperate lakes, however, $CO_2$ is a poor predictor of  photosynthetic fractionation
(Bade et al., 2006). Our lowest modeled fractionation values reflected active uptake of $HCO_3^-$,
supported by elevated phytoplankton isotopic values. In contrast, northern temperate lakes had a
narrower range of phytoplankton isotopic composition (lower on average), and overall higher
ambient $CO_2$ concentrations, both attributable to heterotrophic degradation of terrestrial carbon.
These results suggest an important distinction in carbon cycling between these two regions,
where inorganic carbon availability appears to drive photosynthetic fractionation in eutrophic
lakes, but is likely controlled by other processes (e.g., temperature) in low-nutrient ones.

Phytoplankton stable isotopic composition is dependent on both on the isotopic

composition of DIC source material and fractionation during cellular uptake and assimilation. In
our study, the DIC source material ($\delta^{13}C_{DIC}$) was enriched in $^{13}C$ across all lakes and sampling
events, with values ranging from -12.5 to + 5.8 ‰, within the range of previously measured
values for eutrophic lakes in the same region (de Kluijver et al., 2014). Source values in this
range are likely attributable to dissolution of mineral bicarbonate (Mook 1986; Boutton 1991;
Bade et al. 2004), but could also be sourced from the atmosphere or biogenic methane
production via acetate fermentation (Drimmie et al., 1991; Simpkins and Parkin, 1993; Stiller
and Magaritz, 1974). In northern temperate lakes, $\delta^{13}C_{DIC}$ values are generally lower than those
measured in our study (e.g., < -25 ‰; Bade et al., 2006), attributable to heterotrophic
degradation of terrestrial organic matter (Bade et al., 2007), which is negligible relative to
autochthonous organic matter in the eutrophic surface waters of our study sites (authors'
unpublished data; in review). Collectively, the active uptake by phytoplankton of DIC source
material enriched in $^{13}C$ combined with decreased photosynthetic fractionation due to CCM
processes result in an increase in the carbon stable isotopic signature of the phytoplankton
community.

We found a significant positive relationship between photosynthetic fractionation and

$\delta^{13}C_{DIC}$. Across trophic gradients (i.e,, $\delta^{13}C_{DIC}$ values between -30 ~ + 5 ‰, Bade et al. 2004; de
Kluijver et al. 2014, this study), these relationships are driven by decreases in $\delta^{13}C_{DIC}$ values with
increasing biomass (i.e., blooms), and decreased fractionation as CCMs are induced (Sharkey
and Berry, 1985). Our results suggest that CCMs are functioning and fractionation is lowest
when the DIC pool is enriched in $^{13}C$ (~ -15 to 0 ‰, Boutton 1991). In addition to CCMs, it is
possible that observed decreases in photosynthetic fractionation are attributable in part to
diffusive limitation, i.e., photosynthetic fractionation decreases because $^{12}C$ is depleted from the
water column and predominantly $^{13}C$ remains (Raven et al., 2005). During blooms in these very
productive systems, however, pH consistently exceeds 8.3 (Table 1), making the dominant
inorganic carbon species $HCO_3^-$ due to geochemical carbonate equilibria processes. While rapid
diffusive uptake of atmospheric $CO_2$ near the air-water interface is possible for surface blooms,
an active uptake mechanism (CCM) is necessary for $HCO_3^-$ utilization and to sustain blooms for
weeks to months at a time, as was observed in our study.

Our results have important implications for how cyanobacteria blooms may be sustained

in anthropogenically eutrophic systems. It is well established that high nutrient concentrations
result in high phytoplankton biomass (Heisler et al., 2008). It is less clear, however, what
mechanisms cause variability in timing and duration of blooms among eutrophic and
hypereutrophic lakes. CCMs may provide a competitive advantage to cyanobacteria when high
primary productivity depletes ambient $CO_2$. This mechanism may allow blooms to be sustained
for weeks to months at a time with negligible concentrations of $CO_2$ in the water column
(Cotovicz et al., 2015). While nutrient reduction is ultimately critical in the prevention of blooms
(Heisler et al., 2008; Rigosi et al., 2014), the mechanism presented here provides insight into
causes of bloom duration and intensity at high nutrient concentrations.

Our results show that eutrophic lakes function substantially differently than less impacted

surface waters. Temperate lakes are generally considered sources of $CO_2$ to the atmosphere
(Tranvik et al., 2009). We demonstrate that phytoplankton CCM use allows dense phytoplankton
to grow at low $CO_2$ concentrations and may facilitate extended periods of high primary
production, $CO_2$ depletion, and atmospheric $CO_2$ uptake in surface waters.  These processes may
increase sediment C burial and the export of autochthonous organic C (Heathcote and Downing,
2011; Pacheco et al., 2014), and may have the potential to increase methane emissions from
anoxic sediments (Hollander and Smith, 2001). Our work demonstrates fundamental differences
in inorganic carbon utilization between northern temperate and agricultural, eutrophic lakes.
Because the extent of impacted, high nutrient lakes is predicted to  increase with the food
demands of a growing human population (Foley et al., 2005), understanding mechanisms driving
carbon cycling in these systems will be critical in evaluating the impact of cyanobacteria blooms
on global carbon cycles.

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

**Author contributions** AMMW and JAD jointly conceived the study. AMMW wrote the
manuscript, conducted field sampling and laboratory analysis, and analyzed data. ADW
contributed stable isotope methodology and laboratory analyses. JAD supervised the project. The
authors declare no competing interests.
**Acknowledgements** We thank Amber Erickson, Lisa Whitehouse, Dan Kendall, Clayton
Williams, and Suzanne Ankerstjerne for chemical and analytical assistance, James Cotner and
James Raich for comments on early versions of the manuscript, and Adam Heathcote for his
contributions to site selection and sampling design. Thank you to Drs. McConaughey,
Verspagen, and one anonymous reviewer for constructive comments on the manuscript. This
study was funded by a grant from the National Science Foundation to John A. Downing, DEB-

517  1021525.

**Figure legends**
**Figures 1-2.** Community composition (division level) and biomass for three summer sampling
points in each lake.
**Figure 3.** Correlation between phytoplankton $\delta^{13}C$ and chlorophyll $a$, indicating isotopic
enrichment increased with phytoplankton biomass. Dashed line indicates phytoplankton bloom
conditions, defined here as >40 µg Chl $a$ $L^{-1}$ (Bachmann et al., 2003).
**Figure 4. Top.**  Non-linear relationship between the stable isotopic ambient $pCO_2$ concentration
in surface water and the stable carbon isotopic signature of the phytoplankton community.
**Bottom**. Non-linear relationship between photosynthetic fractionation ($\varepsilon p$, biomass relative to
ambient $CO_2$) and $pCO_2$. The vertical line indicates atmospheric equilibrium when samples were
collected (393 ppm). Color of points indicates Chl a concentration: white = 0-40 µg Chl $a$ $L^{-1}$;
grey = 41- 100 µg Chl $a$ $L^{-1}$; black= > 100 µg Chl $a$ $L^{-1}$. Vertical line indicates atmospheric $CO_2$
equilibrium when study was conducted (393 ppm).

**Figure 5.** Linear relationship between the stable isotopic signature of the ambient DIC pool and
photosynthetic carbon fractionation ($\varepsilon_p$, biomass relative to ambient $CO_2$. Color of points
indicates Chl a concentration: white = 0-40 µg Chl $a$ L$^{-1}$; grey = 41- 100 µg Chl $a$ L$^{-1}$; black= >
100 µg Chl $a$ L$^{-1}$.

| Lake | n | Latitude | Longitude | TP ($\mu g\ L^{-1}$) | TN ($mg\ L^{-1}$) | Chl a ($\mu g\ L^{-1}$) | TA ($mg$ $CaCO_3\ L^{-1}$) | pH | $\delta^{13}DIC$ (‰ VPBD) |
|------|---|----------|-----------|---------|---------|---------|---------|-----|--------|
| Arrowhead | 13 | 42.297218 | -95.051228 | 25 ± 8 | 0.8 ± 0.1 | 10 ± 6 | 190 ± 8 | 8.4 ± 0.1 | -1.68 ± 1.08 |
| Badger | 13 | 42.586161 | -94.192562 | 58 ± 35 | 9.4 ± 5.7 | 33 ± 34 | 166 ± 33 | 8.3 ± 0.4 | -2.60 ± 1.96 |
| Beeds | 12 | 42.770320 | -93.236436 | 75 ± 48 | 7.4 ± 4.5 | 48 ± 40 | 193 ± 37 | 8.4 ± 0.3 | -3.12 ± 1.31 |
| Big Spirit | 11 | 43.479377 | -95.083424 | 46 ± 22 | 1.1 ± 0.3 | 22 ± 22 | 168 ± 7 | 8.6 ± 0.1 | 0.51 ± 1.03 |
| Black Hawk | 12 | 42.296334 | -95.029191 | 225 ± 118 | 2.4 ± 0.5 | 78 ± 35 | 188 ± 12 | 8.8 ± 0.2 | 2.61 ± 1.25 |
| Center | 13 | 43.412607 | -95.136293 | 104 ± 50 | 1.8 ± 0.2 | 41 ± 36 | 163 ± 4 | 8.5 ± 0.2 | 2.97 ± 1.70 |
| East Osceola | 11 | 41.032548 | -93.742649 | 195 ± 77 | 1.9 ± 0.4 | 80 ± 47 | 111 ± 27 | 8.8 ± 0.6 | -4.92 ± 2.00 |
| Five Island | 14 | 43.145274 | -94.658204 | 106 ± 50 | 2.1 ± 0.3 | 67 ± 37 | 165 ± 10 | 8.4 ± 0.2 | 2.58 ± 1.48 |
| George Wyth | 13 | 42.534834 | -92.400362 | 62 ± 22 | 1.0 ± 0.2 | 26 ± 7 | 141 ± 26 | 8.4 ± 0.2 | -1.63 ± 1.54 |
| Keomah | 13 | 41.295123 | -92.537482 | 106 ± 105 | 1.4 ± 0.6 | 44 ± 52 | 117 ± 15 | 8.6 ± 0.4 | -4.70 ± 1.44 |
| Orient | 12 | 41.196669 | -94.436084 | 397 ± 286 | 2.3 ± 1.2 | 144 ± 105 | 98 ± 22 | 9.4 ± 0.4 | -5.01 ± 5.36 |
| Lower Gar | 11 | 43.352299 | -95.120186 | 95 ± 35 | 1.6 ± 0.2 | 50 ± 23 | 186 ± 14 | 8.6 ± 0.1 | 0.19 ± 1.59 |
| Rock Creek | 12 | 41.736936 | -92.851859 | 115 ± 44 | 1.7 ± 0.4 | 52 ± 49 | 148 ± 7 | 8.5 ± 0.2 | -1.43 ± 1.64 |
| Silver-D | 12 | 43.439162 | -95.336799 | 161 ± 85 | 2.1 ± 0.9 | 35 ± 58 | 174 ± 17 | 8.4 ± 0.2 | -2.52 ± 1.23 |
| Silver-PA | 12 | 43.030775 | -94.883701 | 339 ± 206 | 2.5 ± 0.6 | 117 ± 60 | 163 ± 32 | 8.8 ± 0.3 | 3.25 ± 1.62 |
| Springbrook | 12 | 41.775930 | -94.466736 | 38 ± 25 | 1.8 ± 0.9 | 17 ± 14 | 181 ± 20 | 8.3 ± 03 | -3.66 ± 1.08 |

Table 1. Summary data for lakes included in this study. Total phosphorus (TP), total nitrogen (TN), chlorophyll a (Chl a), total
alkalinity (TA), pH, and $\delta^{13}DIC$ are reported as average values of all sampling events (ice free season, April to November 2012) ±
standard deviation; n represents the number of observations per lake.


| Lake | n | Chl a (μg L⁻¹) | TA (mg L-1 CaCO3-) | pH | $\delta^{13}DIC$ (‰ VPDB) | $\delta^{13}POC$ (‰VPDB) | $\varepsilon_p$ | pCO2 (ppm) |
|---|---|---|---|---|---|---|---|---|
| Arrowhead | 0 | NA | NA | NA | NA | NA | NA | NA |
| Badger | 4 | 71 ± 20 | 133 ± 28 | 8.7 ± 0.4 | -1.31 ± 1.40 | -25.55 ± 2.66 | 22.70 ± 2.23 | 234 ± 289 |
| Beeds | 4 | 101 ± 49 | 170 ± 40 | 8.6 ± 0.2 | -2.23 ± 1.00 | -24.07 ± 1.52 | 20.28 ± 2.32 | 240 ± 195 |
| Big Spirit | 3 | 68 ± 28 | 168 ± 10 | 8.7 ± 0.1 | 1.43 ± 0.60 | -27.04 ± 1.20 | 26.99 ± 0.83 | 227 ± 29 |
| Black Hawk | 9 | 86 ± 32 | 184 ± 10 | 8.8 ± 0.3 | 2.75 ± 0.91 | -22.34 ± 1.32 | 23.56 ± 1.36 | 221 ± 107 |
| Center | 8 | 73 ± 27 | 164 ± 4 | 8.7 ± 0.2 | 4.11 ± 0.90 | -22.51 ± 1.23 | 25.05 ± 1.01 | 172 ± 92 |
| East Osceola | 9 | 69 ± 24 | 107 ± 26 | 8.9 ± 0.6 | -5.08 ± 2.23 | -24.79 ± 3.55 | 18.07 ± 4.88 | 241 ± 457 |
| Five Island | 10 | 84 ± 32 | 163 ± 9 | 8.4 ± 0.1 | 2.92 ± 1.54 | -24.65 ± 0.98 | 26.23 ± 1.67 | 451 ± 224 |
| George Wyth | 0 | NA | NA | NA | NA | NA | NA | NA |
| Keomah | 4 | 63 ± 22 | 103 ± 11 | 9.0 ± 0.3 | -4.36 ± 1.58 | -24.79 ± 1.57 | 18.53 ± 3.18 | 29 ± 34 |
| Orient | 9 | 175 ± 77 | 90 ± 20 | 9.5 ± 0.5 | -5.80 ± 5.90 | -18.38 ± 3.13 | 10.73 ± 8.33 | 42 ± 53 |
| Lower Gar | 7 | 66 ± 17 | 177 ± 7 | 8.7 ± 0.1 | 1.03 ± 0.87 | -25.84 ± 1.04 | 25.44 ± 0.74 | 293 ± 86 |
| Rock Creek | 7 | 70 ± 19 | 148 ± 8 | 8.6 ± 0.2 | -0.78 ± 1.61 | -25.42 ± 2.08 | 23.19 ± 1.47 | 266 ± 146 |
| Silver-D | 3 | 96 ± 62 | 168 ± 12 | 8.7 ± 0.2 | -0.92 ± 0.91 | -27.65 ± 0.44 | 25.22 ± 0.71 | 208 ± 78 |
| Silver-PA | 11 | 135 ± 69 | 163 ± 34 | 8.8 ± 0.4 | 3.59 ± 1.24 | -24.27 ± 1.90 | 26.32 ± 1.39 | 234 ± 177 |
| Springbrook | 1 | 48 | 174 | 8.0 | -2.50 | -28.57 | 24.71 | 375 |

Table 2. Average chemical conditions during bloom events (Chl a > 40 µg L⁻¹). Values are average ± standard deviation of n observations
occurring when Chl a exceeded 40 µg L⁻¹. Values are not reported for Arrowhead and George Wyth Lakes because Chl a values never exceeded
this threshold.

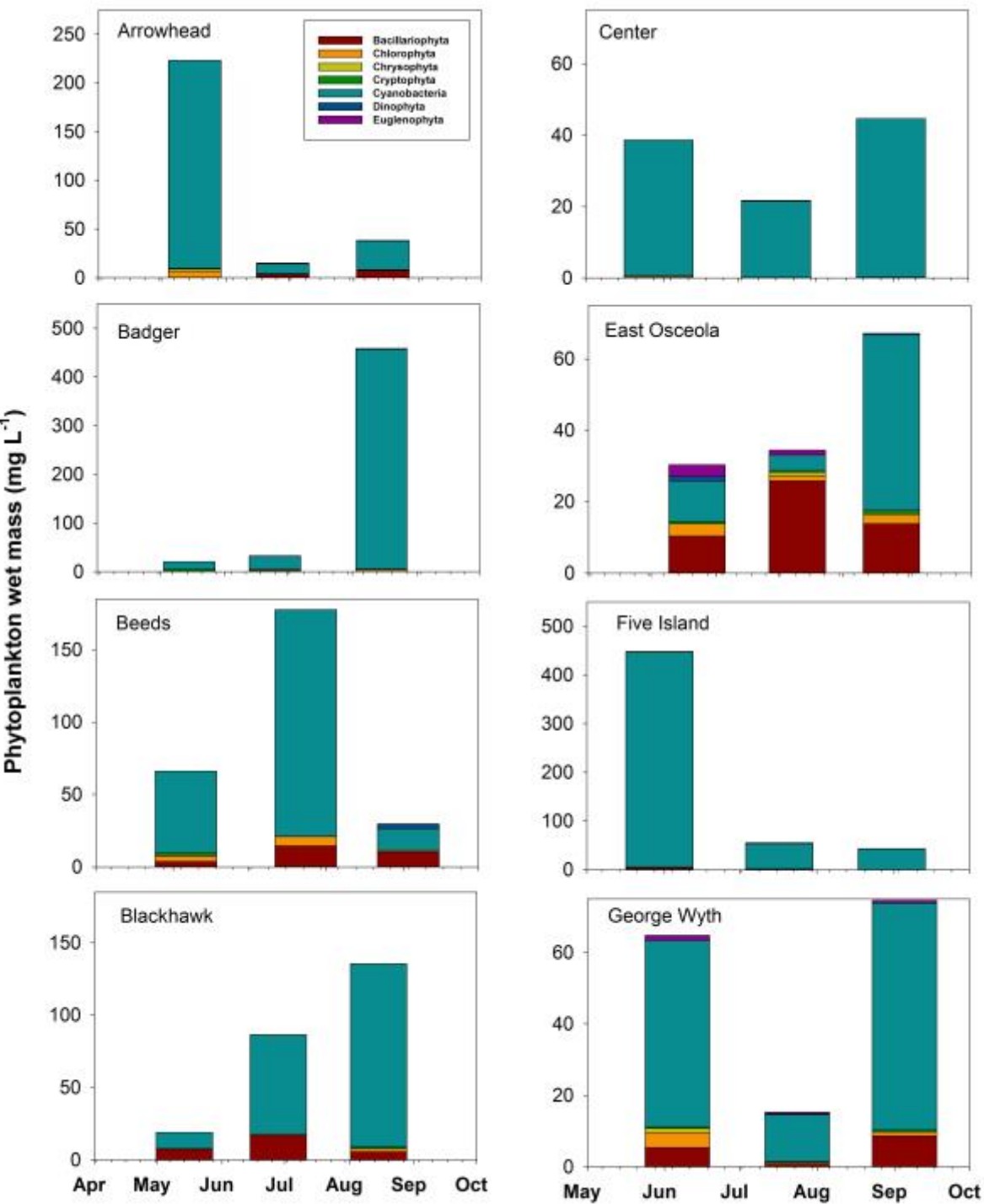


**Figure 1.**

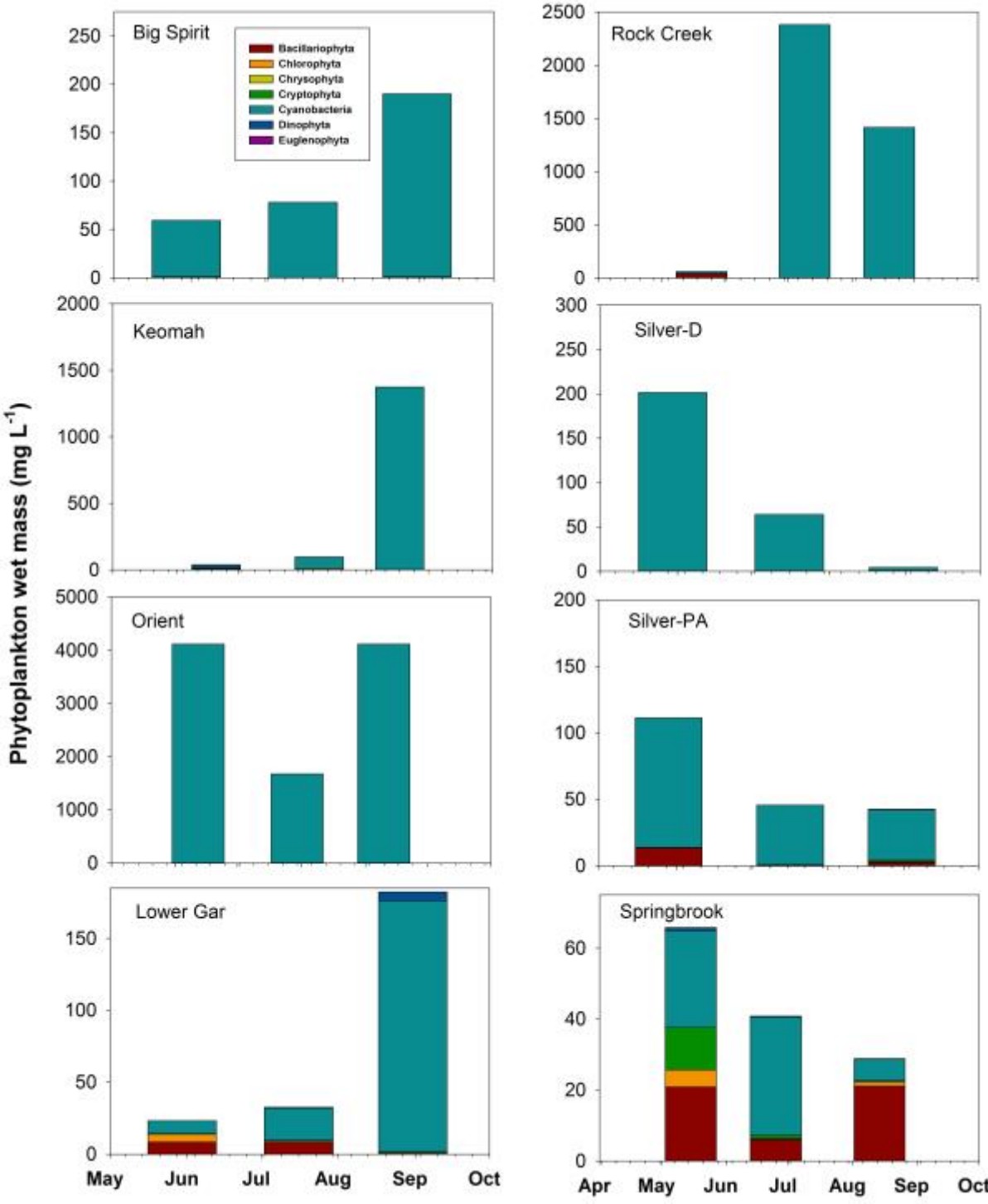


**Figure 2.**

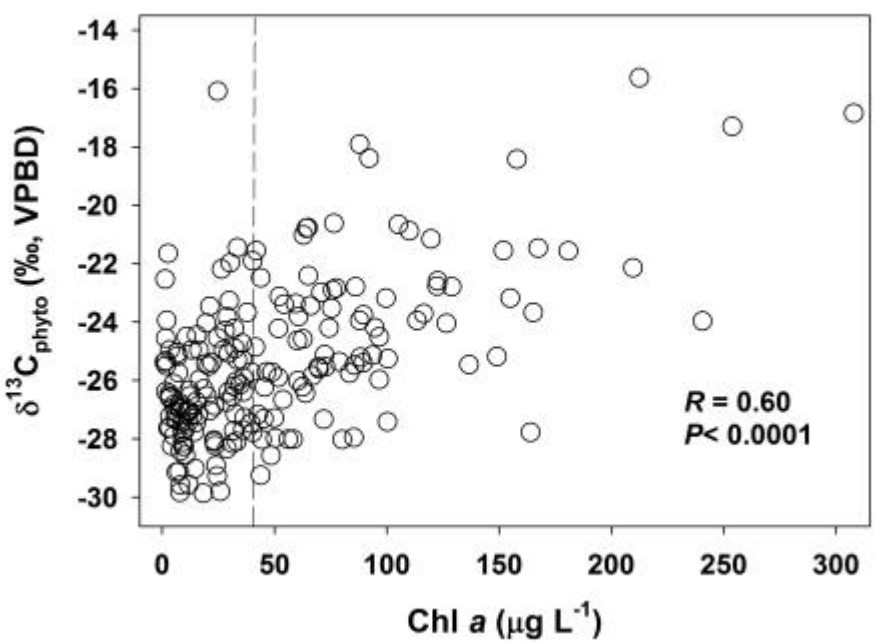


**Figure 3.**







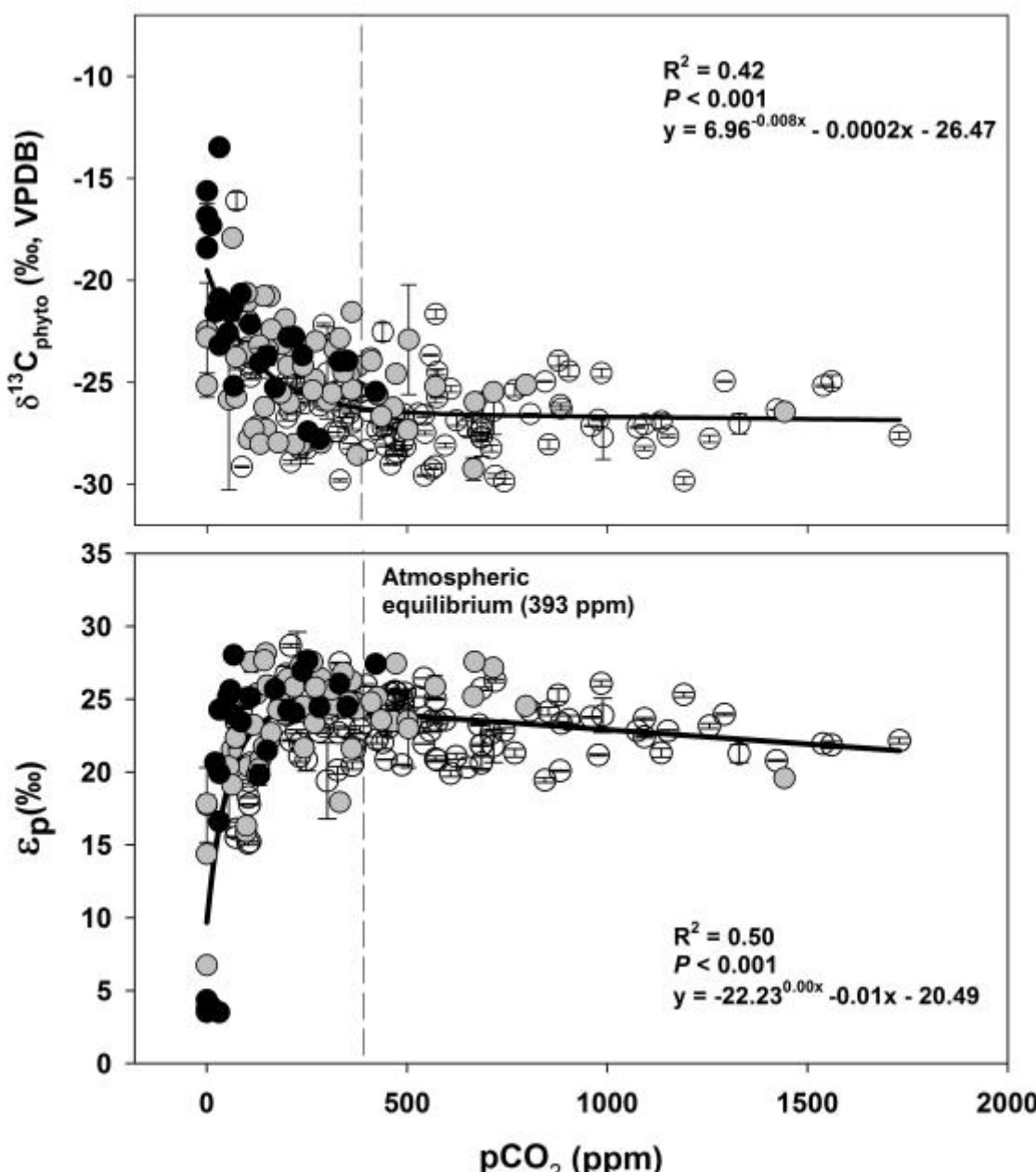


**Figure 4.**

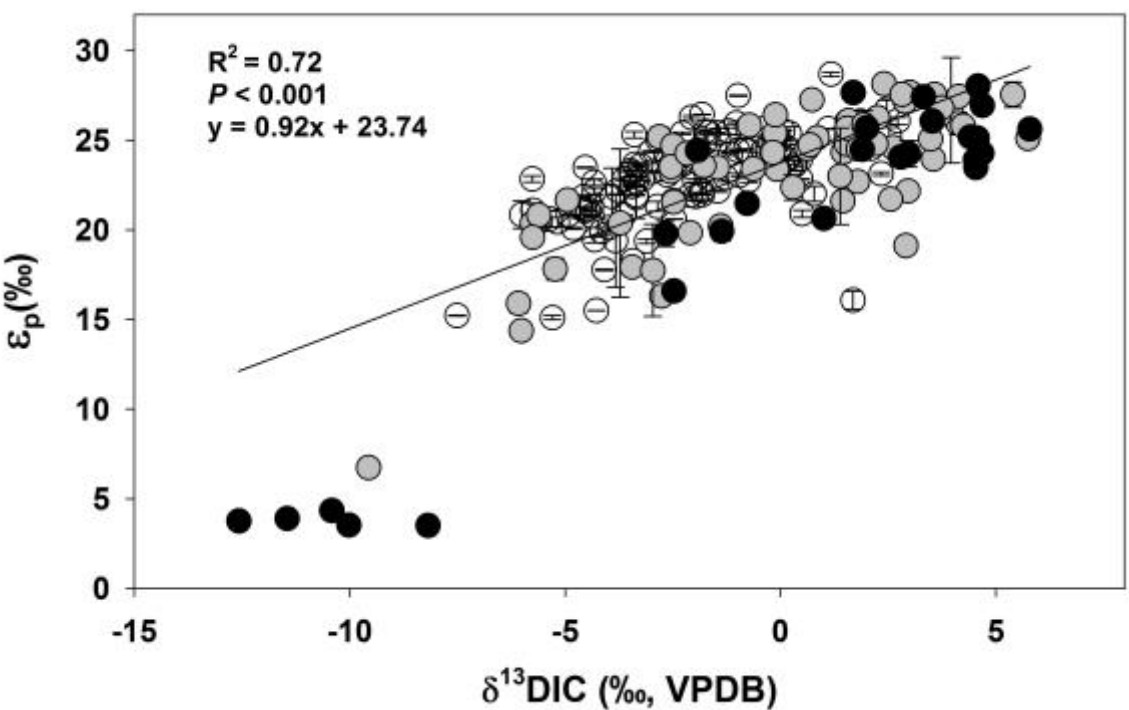


**Figure 5.**