# Peer review of "Cyanobacterial carbon concentrating mechanisms facilitate sustained CO2 depletion in eutrophic lakes Ana M. Morales-Williams1,2,3, Alan D. Wanamaker4, Jr., and John A. Downing1,5 1Department of Ecology, Evolution, and Organismal Biology, Iowa St"

_Biogeosciences, 2016_

## Referee Comment (RC1) · T. McConnaughey (Referee) · 1 Nov 2016

Morales-Williams Bg-2016-350 Morales-Williams et al have assembled an isotopic data base on aquatic plankton and ambient water, and relate this to carbon concentrating mechanisms (CCMs) in cyanobacteria. CCMs are like cosmological "dark energy". One might doubt their existence, except that many smart people confidently attest to them. It is nevertheless ironic that CCMs are reported mainly from very small algae. Small size makes CCMs less necessary, and harder to operate. Biological lipid membranes just don't retain CO2. They retain HCO3- much better, but 1000-fold accumulation factors (Line 72: Raven et al., 2008) make even an acolyte squirm. (Some papers suggest even higher accumulation ratios.) Is this necessary? Isn't it expensive? 1000x

also brings internal DIC levels nearly to the molar range. Wouldn't that create high osmotic pressures, and pop those little bugs? Morales-Williams et al attribute high 13C levels in phytoplankton to CCMs. Do the isotopes really require a CCM? What sorts of CCM would or wouldn't explain the isotopic results, and what can be inferred about the CCM? For example, what does the isotopic data say about internal carbon concentration factors? Leakage rates? Title: "Carbon concentrating mechanisms maintain bloom biomass and CO2 depletion in eutrophic lake ecosystems" doesn't mention cyanobacteria or isotopic measurements, which are the focus of the paper.

Shallow surface water systems are rife with isotopic complications. Wintertime decomposition of organic matter brings springtime high CO2, low pH, low 13C-DIC. Even methane production (line 245) and methane oxidation might alter the 13C of DIC. Hydrology and groundwater inputs can be important (line 244). Carbonate rocks in the soils, like the glacial tills in Iowa, can add isotopically heavy DIC to the system. The present data is further complicated by many different ponds, with individual depths, presence or absence of macrophyte beds, farm water inputs, surface algal scums, different species of algae, blooms at different times, etc. With all this heterogeneity, focus on summertime algal bloom conditions. In the graphs, use larger or darker symbols for bloom conditions. In the table, give separate values typical of bloom conditions, and include representative pH, alkalinity, and chlorophyll. In text, please summarize chemical conditions during algal blooms.

Line 96: "decreased carbon efflux". Carbon efflux may be key to the isotopic balance. Carbon balance models for cyanobacterial CCMs (like Manger and Brennon 2014) sometimes call for large carbon effluxes, sometimes much larger than photosynthetic fluxes. CO2 efflux might leave internal HCO3- relatively enriched in C13, leading to C13 enrichment of photosynthetic products.

159: Phytoplankton samples fumed in HCl to remove inorganic carbon. This procedure would mainly be useful if the samples contained lots of it. Its quantity and isotopic composition would be very nice to know. Could you possibly make such measure-

ments? Could CaCO3 or other solid phases account for some of this internal C? Many cyanobacteria do calcify. Calcification is most likely in alkaline waters with significant calcium. Please list ambient pH and alkalinity levels in table 1, and discuss this possibility. Calcification can also act as a CO2 generator (McConnaughey 2012, Mar Ecol Prog Ser doi: 10.3354/meps09776).

163 "appropriate isotopic scale?"

191 fractionation of biomass compared to external CO2. (Eq 4 line 173): $\varepsilon$p =($\delta$13CCO2 - $\delta$13Cphyto ) / (1 + ($\delta$13Cphyto / 1000)). Text line 191 (as is figure 2 caption) should specify that you are talking about fractionation of biomass relative to ambient CO2 to prevent confusion (for example, confusion with ambient DIC, internal DIC pool, or internal CO2.) Note that this fractionation factor is a result of the cumulative fractionations that have occurred as the plankton grew. It is not an instantaneous fractionation that occurs at the time of harvest, during the bloom. Can you estimate an instantaneous fractionation?

23, 204: "Harmful" and HCB: This may be true from a human or fish perspective, but this study doesn't address harm.

234, 252: Isotopically light aquatic DIC often comes from decomposition of organic matter, especially in early spring, accompanied by high total DIC and low pH. However, CO2 invasion from air and hydroxylation in alkaline waters during summertime bloom, accompanied by kinetic isotope fractionations, might also cause isotopic enlightenment of DIC.

---

## Referee Comment (RC2) · J. Verspagen (Referee) · 23 Nov 2016

General comments:

In this manuscript, Morales et al. present carbon isotope data collected from 16 eutrophic lakes, and show that when dissolved CO2 concentrations become undersaturated, the isotopic composition of particulate organic carbon increases, while the photosynthetic fractionation decreases. These findings are attributed to the (increased) use of Carbon Concentrating Mechanisms (CCMs) by phytoplankton (i.e., the ability to utilize bicarbonate as an inorganic carbon source) when dissolved CO2 concentrations are depleted. These findings are not entirely novel, similar relations are described in Smyntek et al (2012), however, the data by Morales et al show that these findings apply

to a wide(r) range of lakes.

I have two major concerns (detailed below): 1) There is a strong emphasis on cyanobacteria and cyanobacterial blooms in the Introduction section, which is not reflected by the results section, in which only chlorophyll a concentrations are shown. The authors should either reduce the emphasis on cyanobacterial blooms in the Introduction section, or proof that the blooms they sampled were dominated by cyanobacteria. 2) I have a problem with the use of a nonlinear dynamic regression to fit the patterns in Figs 2-4: these regressions do not test an expected relation. However, in Smyntek et al (2012), an isotopic fractionation model is presented that probably fits the data in Fig. 3 and 4. I recommend to fit the Smyntek model to your data, it would make the results much stronger.

Specific comments:

The title suggests that CCMs maintain (phytoplankton) bloom biomass. Yet, no evidence is presented that shows a direct relation between CCM activity (i.e. photosynthetic fractionation or delta 13 POC values) and phytoplankton biomass, and no evidence is presented that the use of CCMs maintain phytoplankton biomass.

In the Introduction section and in the Discussion section, there is a strong emphasis on cyanobacteria and cyanobacterial blooms. Yet, in the title, the material and methods section, and the results section, there is no mention of cyanobacterial blooms, only of phytoplankton blooms and/or phytoplankton biomass. Are the blooms that you sampled cyanobacterial blooms? Do you have any information on the bloom composition in the lakes you sampled?

Line 70, and lines 259-260: It is assumed here that eukaryotic CCMs are, by definition, less efficient than cyanobacterial CCMs. I'm not convinced. Firstly, recent research suggests that the key components of eukayotic CCMs (although not fully resolved) are very similar to cyanobacterial CCMs (Moroney and Ynalvez 2007, Wang et al 2011, Meyer and Griffiths 2013). Secondly, there is experimental evidence that some chlorophytes can outcompete cyanobacteria at low CO2 concentrations, even when these cyanobacteria have a complete CCM (i.e. they have all known bicarbonate uptake systems). For competition experiments between a cyanobacterium and a chlorophyte, see Verschoor et al (2013) and Li et al (2016), for cyanobacterial CCM gene composition of Synechocystis PCC 6803, see Price et al (2008).

Lines 93-104: In this section the authors suggest that cyanobacteria that use CCMs to take up bicarbonate have elevated delta 13C signatures: how about the delta 13C signature of eukaryotic phytoplankton (particularly chlorophytes) that use a CCM to take up bicarbonate? According to the references in lines 215-216, marine eukaryotic phytoplankton also have elevated delta 13C signatures.

Line 113: "16 lakes were chosen based on . . . survey data". What were the selection criteria?

Line 120-124: Here a listing is given of standard physical, chemical and biological parameters measured at each sampling event. Many of these parameters are not referred to in the results section. Please remove these parameters from the text, or present and discuss them in the results/discussion section. Also, please add alkalinity and pH to Table 1.

Lines 171-173 (equations 2-4). Please explain the parameters in these equations, e.g. in particular, what do epsilon(a) and epsilon(b) mean?

I have some concerns about the statistical analysis of the dataset. 1) I wonder whether one has to control for the different lakes. The reason for my concern is that the shape of the fits of the nonlinear regressions of Figs 2, 3 and 4 rely heavily on 6-7 points at low pCO2/low photosynthetic fractionation/low delta 13C of POC. Note that low delta 13C of POC does not necessarily imply high chl a concentrations (Fig. 1). These 6-7 points might come from 1 outlier lake. For this reason, I'm not sure whether a nonlinear dynamic regression (as presented in Figs 2-4) is an appropriate statistical procedure to analyze the dataset. If I understand correctly, nonlinear dynamic regression is an

iterative process that may converge to find the best possible curve that fits the dataset. It does not test an expected relation between a dependent and an independent parameter. In Smyntek et al (2012), an isotopic fractionation model is presented (in Eqs 1 and 2, plotted in Fig. 2 of Smyntek et al 2012) that shows relations between pCO2 and delta 13C of POC, and between pCO2 and the photosynthetic fractionation that look remarkably similar to the shape of the curves that were derived in this study by nonlinear dynamic regression (i.e. Fig. 3 and 4). The Smyntek model should also predict the relation between delta 13DIC and the photosynthetic fractionation in Fig. 2. It makes perfect sense to test whether the fractionation model by Smyntek et al (2012) fits your dataset.

Line 198-199: what kind of regressions are given here? Linear regressions of data with a pCO2 < 393? Please be more precise: give the name of the regression and the statistical parameters: e.g. Linear regression, $R^2$ = 0.90, P < 0.01, N = 10

Technical corrections:

Table 1: please add two extra columns, one with the averaged alkalinity, and one with the number of observations per lake (N).

Fig. 1: x-axis label should be "Chl a (ug L-1)"

References:

Li, W., Xu, X., Fujibayashi, M., Niu, Q., Tanaka, N., & Nishimura, O. (2016). Response of microalgae to elevated CO2 and temperature: impact of climate change on freshwater ecosystems. Environmental Science and Pollution Research, 23(19), 19847-19860.

Meyer, M., & Griffiths, H. (2013). Origins and diversity of eukaryotic CO2-concentrating mechanisms: lessons for the future. Journal of experimental botany, 64(3), 769-786.

Moroney, J. V., & Ynalvez, R. A. (2007). Proposed carbon dioxide concentrating mechanism in Chlamydomonas reinhardtii. Eukaryotic cell, 6(8), 1251-1259.

Price, G. D., Badger, M. R., Woodger, F. J., & Long, B. M. (2008). Advances in understanding the cyanobacterial $CO_2$-concentrating-mechanism (CCM): functional components, Ci transporters, diversity, genetic regulation and prospects for engineering into plants. Journal of experimental botany, 59(7), 1441-1461.

Verschoor, A. M., Van Dijk, M. A., Huisman, J., & Van Donk, E. (2013). Elevated $CO_2$ concentrations affect the elemental stoichiometry and species composition of an experimental phytoplankton community. Freshwater Biology, 58(3), 597-611.

Wang, Y., Duanmu, D., & Spalding, M. H. (2011). Carbon dioxide concentrating mechanism in Chlamydomonas reinhardtii: inorganic carbon transport and $CO_2$ recapture. Photosynthesis research, 109(1-3), 115-122.

---

## Referee Comment (RC3) · Anonymous Referee #3 · 7 Dec 2016

Review of Morales-Williams – CCMs maintain bloom biomass and CO2 depletion in eutrophic lake systems

The MS by Morales-Williams et al shows carbon isotope data from 16 different lakes (eutrophic and hypereutrophic). The carbon fractionation factors were calculated and correlated to the CO2 concentration available. The authors suggest that the decrease in fractionation is due to the use of HCO3-, indicating CCM activity, which would allow the phytoplankton community to thrive in the lakes even when CO2 becomes limited.

I have several major concerns with the data presentation:

- A simple correlation of d13C values with Chl a concentration cannot be used in this

study to predict CCM activity. The authors describe the function of the CCM and how this could potentially change the isotopic signature of the cells (see line 91). Recent papers by Eichner et al 2015 and Raven and Beardal 2015 include internal cycling and loss terms of CO2. These two paper directly affect the interpretation of the data in this MS and should be introduced and discussed. Additional, a paper by Kranz et al 2015 showed the change in epsilon 13C during a bloom of diatoms. These authors also measured CCM parameters directly, seeing a switch from CO2 to HCO3- uptake at low CO2 conditions. However, this study used a model (Hopkinson) to predict the changes in d13C POC due to the switch to HCO3- uptake. The authors could contribute less than 0.5 permill change in the d13C signal to the switch in the inorganic carbon source- . Together with the findings by Eichner et al 2015 and Raven and Beardall (2015) I feel that the authors have be aware that isotopic signal of organic matter are not necessarily driven by the uptake of different carbon species, but largely are affected by other cellular processes such as leakage as well as the external d13C DIC. Additionally, different species have different isotopic compositions – do the authors know if the lakes have similar phytoplankton communities?

- In the method section the authors do not specifically mention how they obtained the biomass measured. Please be more precise in this and also mention how much of the organic material might have been detritus from other sources.

- The authors have to include the data of TA, DIC, d13C DIC, pH into Table 1 for the reader to understand the dataset and the correlations given.

- The title of the MS is a little farfetched. Neither does the study proof that CCMs maintain biomass in the lakes not did the authors show actual CCM activity. Please revise.

Specific comments:

Line 113: What are the criteria for which the lakes have been chosen? Wouldn't it be sufficient to just mention that 16 lakes were sampled and then briefly describe their

properties?

Line 127,132: I don't understand the expression – "measurements were increased" do the authors mean that data was corrected/adjusted for pressure and temperature? Please revise.

Line 143-145. I feel that this short paragraph should move behind line 160.

Line 171 and 172: describe what alpha a and alpha b means (Temperature-dependent fractionation factors between CO2 and HCO3 (a) as well as HCO3 − and CO32− (b).

Add additional info on the sampling of the phytoplankton organic matter

Fig 1: Despite being significant, the predictive power of the dataset is relatively low! How would the dataset look like, if you use epsilon vs. Chl a. I feel that this would be more appropriate especially after reading how d13C seems to change in the different lakes.

Discussion:

Line 220: Please rephrase: "This mechanism likely provides a competitive..." The authors refer to decreased fractionation as a mechanism, yet the fractionation calculated is the result of cellular mechanisms such as enhanced HCO3- uptake and/or enhanced CO2 leakage. Maybe rephrase to: "The cellular mechanisms which led to the decrease in fractionation under low pCO2 likely provide..."

Please explain the paragraph starting line 234 better.

References:

Eichner M Thoms S Kranz SA Rost B . 2015. Cellular inorganic carbon fluxes in Trichodesmium: a combined approach using measurements and modelling. Journal of Experimental Botany 66, 749–759.

Kranz S Young JN Goldman J Tortell PD Bender M Morel FMM . 2015. Low tem-

perature reduces the energetic requirement for the CO2 concentrating mechanism in diatoms. New Phytologist 205, 192–201.

Raven JA Beardall J . 2016. The ins and outs of CO2 . Journal of Experimental Botany 67, 1–13.

---

## Author Comment (AC1) · 13 Jan 2017

Thank you to Drs. McConaughey, Verspagen, and one anonymous referee for their thorough and constructive critiques. Responses to each review are below.

Reviewer 1.

Response to general comments:

Reviewer 1 expresses skepticism regarding the existence of CCMs, comparing them to cosmological "dark energy", and noting the irony of their reports primarily from small phytoplankton. There is a substantial body of literature documenting this system, it's molecular components, and evolutionary origins, which are summarized in our introduction and several reviews on the topic (e.g., Badger et al., 2006; Hopkinson et al., 2016; Kaplan and Reinhold, 1999; Raven et al., 2008). Many of the reports of this mechanism are from small phytoplankton because much of the early work on this topic was done in culture using model organisms. It has, however, been demonstrated in larger diatoms (Trimborn et al., 2009) and larger colonial cyanobacteria (Eichner et al., 2015).

Reviewer 1 asks if CCMs are necessary and whether they are energetically expensive. CCMs are not just the accumulation of inorganic carbon, but collectively the (1) active transport of $CO_2$ or $HCO_3-$ across the cell membrane, (2) partitioning of Rubisco into carboxysomes, and (3) elevation of $CO_2$ around these enzyme complexes (Price et al., 2008). Yes, this is an energetically costly process, but it is necessary in that it both increases photosynthetic efficiency and local bioavailability of inorganic carbon when $CO_2$ is depleted (e.g., rapid growth and high biomass during blooms). When water column pH exceeds 8, $CO_2$ is negligible and $HCO_3-$ is the dominant inorganic carbon species. $HCO_3-$ cannot passively diffuse across phytoplankton cell membranes, and therefore requires some sort of active transport system, which falls under the umbrella of CCMs. We agree that 1000x accumulation of internal inorganic carbon relative to ambient concentration poses an osmotic pressure problem. We do not suggest that our measurements reflect these very high internal concentrations, and are only reporting extreme literature values in this case. We will remove this reference from our introduction if it is contentious.

Reviewer 1 asks if high phytoplankton 13C levels require CCMs, or if this could be attributed to something else. No, high phytoplankton 13C levels alone could be attributable to any process that elevates 13C of their inorganic carbon source. $CO_2$ generated from methanogenic fermentation and carbonate dissolution could cause this. However, we demonstrate that the highest values of phytoplankton 13C correspond to periods of both peak biomass, severe $CO_2$ depletion (approaching 0 ppm), and elevated pH. pH across sites and sampling events ranged from 7.6 +/- 0.2 in the fall to 10.1

+/- 0.1 in July. These data demonstrate that CO2 is both biologically and chemically depleted, and that to support bloom biomass, phytoplankton would need an active uptake mechanism (i.e., part of CCM process) to access HCO3- and convert it to CO2. In addition to the discrete data presented here, we collected continuous pH and temperature sensor data (every 15 minutes) and have calculated time series of CO2 concentration and flux for all sites with discrete alkalinity and conductivity measurements. These data are in review at another journal but can be referenced in a revised version of this manuscript.

Title: We will edit the title to better reflect the focus of the paper.

Reviewer 1 comments on the complicated nature of shallow surface waters and suggests focusing our study only on summertime bloom conditions. We agree that there is considerable heterogeneity among our sampling sites. We feel that information would be lost if we remove data from the shoulder seasons (temporal CO2 depletion and the shift to bloom conditions). We can clarify heterogeneity among sites by analyzing lakes separately with corrections for multiple tests. We will highlight the bloom season as suggested by using larger or darker symbols for bloom conditions in plots.

We will provide separate values of bloom conditions in the table, include representative pH, alkalinity, and chlorophyll a, and summarize chemical conditions during blooms in text as requested.

Specific comments:

Line 96 "decreased carbon efflux": This statement specifically refers to cyanobacterial CCMs, which are less leaky and have lower efflux than eukaryotic CCMs. We will clarify this in the text.

Line 159: Unfortunately, we do not have material that has not been fumed to take these measurements. We do have early trial data collected from 3 of our study sites the year preceding this study that compares fumed and not fumed samples. There

was a measurable but not statistically significant difference between the two – these data can be included for reference if useful. It is true that many marine and benthic cyanobacteria calcify, but this is not as common in eutrophic, freshwater lakes, and has not been observed in our samples.

We will include discrete pH and alkalinity values in Table 1.

Line 163- We will clarify the meaning of "appropriate isotopic scale".

-Line 191- We will specify that we are referring to fractionation of biomass relative to ambient $CO_2$ to prevent confusion. We recognize that this fractionation factor is a result of cumulative fractionations that occurred as the plankton grew, but because their growth and turnover time during a bloom is rapid (on the order of days), we feel that these values adequately reflect fractionation values of interest

Lines 23 and 204. We will remove the word "harmful".

234, 252: We will clarify this statement and include $CO_2$ invasion and hydroxylation in alkaline waters as processes producing lighter DIC.
* * *
Reviewer 2

Response to general comments:

Reviewer 2 major concerns are that (1) emphasis is placed on cyanobacteria, but only chlorophyll data is presented as a measure of bloom biomass, and that (2) nonlinear dynamic regression does not test an expected relation. To address these concerns, we will include phytoplankton community composition data, and test the Smyntek model as suggested.

Response to specific comments:

Chlorophyll a concentration as presented here is commonly used as a metric of phytoplankton biomass. As above, we will edit the manuscript to include community composition and biovolume data (microscope counts).

Regarding emphasis on cyanobacteria: The blooms we sampled were cyanobacteria blooms. We will include these community composition data and update the title, methods, and results accordingly.

Lines 70 and 259-260: Regarding the efficiency of cyanobacterial versus eukaryotic CCMs, we will update our references to include those suggested showing that, in some cases, chlorophytes can outcompete cyanobacteria in culture. However, Price 2008 (as referenced therein) supports our assertion that cyanobacteria are better competitors for inorganic carbon, attributable to the partitioning of Rubisco and elevation of $CO_2$ in carboxysomes not present in eukaryotes, as well as higher Rubisco nitrogen use efficiency and very low levels of photorespiration.

Lines 93-104: Yes, chlorophytes using a CCM would also be expected to have elevated 13C signatures, but the data presented here are cyanobacteria blooms, not chlorophyte blooms. Updating our manuscript to include community composition data will clarify this.

Line 113: Lakes were chosen along an orthogonal gradient of inter-annual variability in cyanobacteria dominance and watershed permeability. We will update the manuscript to include this information.

Line 129-124: We will remove variables not essential to our results from this section, and update Table 1 to include alkalinity and pH.

Lines 171-173: a and b represent temperature-dependent fractionation factors between $CO_2$ and $HCO_3^-$, and $HCO_3^-$ and $CO_3^{2-}$, respectively. We will update the manuscript to include this information.

Statistical analysis: Reviewer 1 also noted that heterogeneity between lakes may complicate our results. Based on these comments, we will partition the data to highlight

effects of individual lakes, and fit the model presented in Smyntek et al. (2012) rather than using dynamic regression.

Lines 198-199: Non-linear dynamic regression

Technical corrections: We will edit Table 1 as suggested and will fix the units on the Figure 1 axis label.
* * *
Reviewer 3

Response to general comments:

The relationship presented in Figure 1 was not meant to predict CCM activity, but rather illustrate the correlation between an increase in biomass and elevated phytoplankton 13C. We will remove the trendline and report R rather than R2 here.

We will update our references to include those suggested here to better treat the issue of leakage. We are aware that the isotopic signal is influenced by leakage and the external d13C DIC. We measured the external d13C and have reported these data.

Regarding composition of the lake phytoplankton communities, as described in our response to Reviewer 2, we will include phytoplankton community composition data.

Reviewer 3 indicates that we do not specifically mention how we obtained the biomass measured. As presented, we have used chlorophyll a as a metric of phytoplankton biomass, which is detailed in the methods. However, when community composition data are included, this can be updated to reflect biovolume measurements from microscopy. Regarding detritus and organic matter from other sources, this is detailed in lines 156-158 of the manuscript.

Table 1: We will make these corrections to include alkalinity and pH, as also suggested by Reviewers 1 and 2. We did not include 13C DIC isotopic data in tabular form because it is presented in Figure 2.

Title: We will revise the title accordingly.

Response to specific comments:

Line 113: The criteria for choosing lakes is described above in our response to Reviewer 2. We feel that including evidence that sample sites were chosen in an informed way and not arbitrarily is useful and should remain in the manuscript.

Lines 127,132: Yes, measurements were corrected for temperature and pressure as described in lines 123-133.

Lines 143-145: We will move this paragraph as suggested.

Lines 171 and 172: Addressed in response to Reviewer 2.

Figure 1: Addressed above in general comments.

Line 220: We will rephrase as suggested.

Paragraph beginning at line 234: Specific suggestions as to what should be clarified would be helpful here, but we will attempt to clarify and better explain these discussion points.

References

Badger, M. R., Price, G. D., Long, B. M. and Woodger, F. J.: The environmental plasticity and ecological genomics of the cyanobacterial $CO_2$ concentrating mechanism., J. Exp. Bot., 57(2), 249–65, doi:10.1093/jxb/eri286, 2006. Eichner, M., Thoms, S., Kranz, S. A. and Rost, B.: Cellular inorganic carbon fluxes in Trichodesmium: A combined approach using measurements and modelling, J. Exp. Bot., 66(3), 749–759, doi:10.1093/jxb/eru427, 2015. Hopkinson, B. M., Dupont, C. L. and Matsuda, Y.: The physiology and genetics of $CO_2$ concentrating mechanisms in model diatoms, Curr. Opin. Plant Biol., 31, 51–57, doi:10.1016/j.pbi.2016.03.013, 2016. Kaplan, A. and Reinhold, L.: Co 2 Concentrating Mechanisms in Microorganisms, 1999. Price, G. D., Badger, M. R., Woodger, F. J. and Long, B. M.: Advances in understanding

the cyanobacterial CO2-concentrating-mechanism (CCM): functional components, Ci transporters, diversity, genetic regulation and prospects for engineering into plants., J. Exp. Bot., 59(7), 1441–61, doi:10.1093/jxb/erm112, 2008. Raven, J. a, Cockell, C. S. and De La Rocha, C. L.: The evolution of inorganic carbon concentrating mechanisms in photosynthesis., Philos. Trans. R. Soc. Lond. B. Biol. Sci., 363(1504), 2641–50, doi:10.1098/rstb.2008.0020, 2008. Trimborn, S., Wolf-Gladrow, D., Richter, K.-U. and Rost, B.: The effect of pCO2 on carbon acquisition and intracellular assimilation in four marine diatoms, J. Exp. Mar. Bio. Ecol., 376(1), 26–36, doi:10.1016/j.jembe.2009.05.017, 2009.

---

## Author Response (AR1)

Thank you to Drs. McConaughey, Verspagen, and one anonymous referee for their thorough and constructive critiques. Responses to each review and relevant changes to the manuscript are detailed below. Reviews below are indented in italics, responses are in normal font. Corresponding changing are highlighted in yellow in the attached manuscript.

**Reviewer 1.**

1. *Biological lipid membranes just don't retain CO2. They retain HCO3- much better, but 1000-fold accumulation factors (Line 72: Raven et al., 2008) make even an acolyte squirm. (Some papers suggest even higher accumulation ratios.) Is this necessary? Isn't it expensive?*

We agree that 1000x accumulation of internal inorganic carbon relative to ambient concentration poses an osmotic pressure problem. We do not suggest that our measurements reflect these very high internal concentrations, and were only reporting extreme literature values in this case. We have removed this statement and reference.

2. *What sorts of CCM would or wouldn't explain the isotopic results, and what can be inferred about the CCM? For example, what does the isotopic data say about internal carbon concentration factors? Leakage rates?*

L80-86. We have included a discussion of structures specific to cyanobacteria CCMs (carboxysomes) that are thought to decrease leakage via a protein shell and thus decrease isotopic discrimination and fractionation, resulting in an intracellular isotopic composition approaching that of source material.

3. *Title: "Carbon concentrating mechanisms maintain bloom biomass and CO2 depletion in eutrophic lake ecosystems" doesn't mention cyanobacteria or isotopic measurements, which are the focus of the paper.*

We have edited our title to include cyanobacteria, but did not include stable isotopes as they are only the methodology we used to characterize the system: "Cyanobacteria carbon concentrating mechanisms facilitate sustained $CO_2$ depletion in eutrophic lakes".

4. *Shallow surface water systems are rife with isotopic complications. Wintertime decomposition of organic matter brings springtime high CO2, low pH, low 13C-DIC. Even methane production (line 245) and methane oxidation might alter the 13C of DIC. Hydrology and groundwater inputs can be important (line 244). Carbonate rocks in the soils, like the glacial tills in Iowa, can add isotopically heavy DIC to the system. The present data is further complicated by many different ponds, with individual depths, presence or absence of macrophyte beds, farm water inputs, surface algal scums, different species of algae, blooms at different times, etc. With all this heterogeneity, focus on summertime algal bloom conditions. In the graphs, use larger or darker symbols for bloom conditions. In the table, give separate values typical of bloom conditions, and*

*include representative pH, alkalinity, and chlorophyll. In text, please summarize chemical conditions during algal blooms.*

We have included a table (Table 2) of average lake chemistry for each site during bloom conditions, defined as chlorophyll *a* exceeding 40 ug/L. A discussion of these data is now included in the results section text at L216-219. We have edited all figures to color points by chl a concentration (white =0-40 ug/L; gray = 41-100 ug/L; black = >100 ug/L, where all values greater than 40 ug/L indicate bloom conditions).

5. *Line 96: "decreased carbon efflux". Carbon efflux may be key to the isotopic balance. Carbon balance models for cyanobacterial CCMs (like Manger and Brennon 2014) sometimes call for large carbon effluxes, sometimes much larger than photosynthetic fluxes. CO2 efflux might leave internal HCO3- relatively enriched in C13, leading to C13 enrichment of photosynthetic products.*

We have addressed the potential for carbon efflux and referenced Manger and Brennon at L80-86.

6. *159: Phytoplankton samples fumed in HCl to remove inorganic carbon. This procedure would mainly be useful if the samples contained lots of it. Its quantity and isotopic composition would be very nice to know. Could you possibly make such measurements? Could CaCO3 or other solid phases account for some of this internal C? Many cyanobacteria do calcify. Calcification is most likely in alkaline waters with significant calcium. Please list ambient pH and alkalinity levels in table 1, and discuss this possibility. Calcification can also act as a CO2 generator (McConnaughey 2012, Mar Ecol Prog Ser doi: 10.3354/meps09776).*

While calcification is common in marine phytoplankton, it is relatively uncommon in eutrophic lakes and was not observed in our study. We have commented on this in the methods section at 181-182. Unfortunately, we are unable to make measurements of the inorganic carbon that was removed in the fuming process. We have included the requested data in Table 1.

7. *163 "appropriate isotopic scale?"*

We have clarified that this refers specifically to VPDB for carbonates at line 185.

8. *191 fractionation of biomass compared to external CO2. (Eq 4 line 173): "p=(13CCO2 - 13Cphyto ) / (1 + (13Cphyto / 1000)). Text line 191 (as is figure 2 caption) should specify that you are talking about fractionation of biomass relative to ambient CO2 to prevent confusion (for example, confusion with ambient DIC, internal DIC pool, or internal CO2.) Note that this fractionation factor is a result of the cumulative fractionations that have occurred as the plankton grew. It is not an instantaneous fractionation that occurs at the time of harvest, during the bloom. Can you estimate an instantaneous fractionation?*

We have edited the text and figure caption, now L188 and 191, and Figures 4 and 5 to specify that we are talking about fractionation of biomass relative to ambient CO2.

9.  *23, 204: "Harmful" and HCB: This may be true from a human or fish perspective, but this study doesn't address harm.*

We have removed the term "harmful" and all instances of the acronym "HCB" from the manuscript and replaced with "cyanobacteria bloom" throughout.

10. *234, 252: Isotopically light aquatic DIC often comes from decomposition of organic matter, especially in early spring, accompanied by high total DIC and low pH. However, CO2 invasion from air and hydroxylation in alkaline waters during summertime bloom, accompanied by kinetic isotope fractionations, might also cause isotopic enlightenment of DIC.*

We have clarified in the text (now 262-263) that these processes may occur in alkaline waters.

**Reviewer 2.**

11. *I have two major concerns (detailed below): 1) There is a strong emphasis on cyanobacteria and cyanobacterial blooms in the Introduction section, which is not reflected by the results section, in which only chlorophyll a concentrations are shown. The authors should either reduce the emphasis on cyanobacterial blooms in the Introduction section, or proof that the blooms they sampled were dominated by cyanobacteria.*

We have included community composition and phytoplankton biomass data (Figures 1 and 2, text L208-213.

12. *2) I have a problem with the use of a nonlinear dynamic regression to fit the patterns in Figs 2-4: these regressions do not test an expected relation. However, in Smyntek et al (2012), an isotopic fractionation model is presented that probably fits the data in Fig.3 and 4. I recommend to fit the Smyntek model to your data, it would make the results much stronger.*

After consideration, we do not feel that we have data appropriate to fit all parameters of the Smyntek model. Fitting the model would require several assumptions that we feel would weaken rather than strengthen our results. The purpose of the dynamic regression in our study is to demonstrate the sharp change point and change in the slope of these relationships with the depletion of CO2. Our results using this approach do, however, closely resemble the best fit of the Smyntek model.

13. *The title suggests that CCMs maintain (phytoplankton) bloom biomass. Yet, no evidence is presented that shows a direct relation between CCM activity (i.e. photosynthetic fractionation or delta 13 POC values) and phytoplankton biomass, and no evidence is presented that the use of CCMs maintain phytoplankton biomass. In the Introduction section and in the Discussion section, there is a strong emphasis on cyanobacteria and cyanobacterial blooms. Yet, in the title, the material and methods section, and the results section, there is no mention of cyanobacterial blooms, only of phytoplankton blooms and/or phytoplankton biomass. Are the blooms that you sampled cyanobacterial blooms? Do you have any information on the bloom composition in the lakes you sampled?*

Yes, these blooms are consistently dominated by cyanobacteria. As above, biomass and community composition data have been added to the manuscript. The title has been edited to include cyanobacteria as suggested by Reviewer 1.

14. *Line 70, and lines 259-260: It is assumed here that eukaryotic CCMs are, by definition, less efficient than cyanobacterial CCMs. I'm not convinced. Firstly, recent research suggests that the key components of eukayotic CCMs (although not fully resolved) are very similar to cyanobacterial CCMs (Moroney and Ynalvez 2007, Wang et al 2011, Meyer and Griffiths 2013). Secondly, there is experimental evidence that some chlorophytes can outcompete cyanobacteria at low CO2 concentrations, even when these cyanobacteria have a complete CCM (i.e. they have all known bicarbonate uptake systems). For competition experiments between a cyanobacterium and a chlorophyte, see Verschoor et al (2013) and Li et al (2016), for cyanobacterial CCM gene composition of Synechocystis PCC 6803, see Price et al (2008).*

We have included a discussion of this uncertainty in the Introduction at 74-77, as well as a discussion of carboxysomes structures unique to the cyanobacteria CCM that are thought to decrease leakage and provide efficiency relative to eukaryotic CCMs (L88-86).

15. *Lines 93-104: In this section the authors suggest that cyanobacteria that use CCMs to take up bicarbonate have elevated delta 13C signatures: how about the delta 13C signature of eukaryotic phytoplankton (particularly chlorophytes) that use a CCM to take up bicarbonate? According to the references in lines 215-216, marine eukaryotic phytoplankton also have elevated delta 13C signatures.*

We fully agree with this statement and did not intend to imply that only cyanobacteria would be isotopically heavier with CCM utilization. The focus was on cyanobacteria in this section because our systems specifically had cyanobacteria blooms, not eukaryotic blooms. We have clarified this in the text at L106.

16. *Line 113: "16 lakes were chosen based on . . . survey data". What were the selection criteria?*

Lakes were chosen along an orthogonal gradient of interannual variability in cyanobacteria dominance and watershed permeability. This has been clarified in the text at L129-131.

17. *Line 120-124: Here a listing is given of standard physical, chemical and biological parameters measured at each sampling event. Many of these parameters are not referred to in the results section. Please remove these parameters from the text, or present and discuss them in the results/discussion section. Also, please add alkalinity and pH to Table 1.*

Alkalinity and pH have been added to Table 1. We have removed mention of meteorological data and depth profiles that were not discussed in the results.

18. *Lines 171-173 (equations 2-4). Please explain the parameters in these equations, e.g. in particular, what do epsilon(a) and epsilon(b) mean?*

These are temperature dependent fractionation factors. This has been clarified at 198-199.

19. *I have some concerns about the statistical analysis of the dataset. 1) I wonder whether one has to control for the different lakes. The reason for my concern is that the shape of the fits of the nonlinear regressions of Figs 2, 3 and 4 rely heavily on 6-7 points at low pCO2/low photosynthetic fractionation/low delta 13C of POC.*

We have addressed this by 1) providing plots of the individual lake relationships for del13C-POC and pCO2 in Supplemental Information. Additionally, we have binned the points in each plot by chlorophyll a concentration as suggested by Reviewer 1.

20. *Note that low delta 13C of POC does not necessarily imply high chl a concentrations (Fig. 1). These 6-7 points might come from 1 outlier lake. For this reason, I'm not sure whether a nonlinear dynamic regression (as presented in Figs 2-4) is an appropriate statistical procedure to analyze the dataset. If I understand correctly, nonlinear dynamic regression is an iterative process that may converge to find the best possible curve that fits the dataset. It does not test an expected relation between a dependent and an independent parameter. In Smyntek et al (2012), an isotopic fractionation model is presented (in Eqs 1 and 2, plotted in Fig. 2 of Smyntek et al 2012) that shows relations between pCO2 and delta 13C of POC, and between pCO2 and the photosynthetic fractionation that look remarkably similar to the shape of the curves that were derived in this study by nonlinear dynamic regression (i.e. Fig. 3 and 4). The Smyntek model should also predict the relation between delta 13DIC and the photosynthetic fractionation in Fig. 2. It makes perfect sense to test whether the fractionation model by Smyntek et al (2012) fits your dataset.*

Please see comment above regarding the Smyntek model. Regarding the iterative fit process, the Smyntek model would also be an iterative process resulting in the best fit for the data.

21. *Line 198-199: what kind of regressions are given here? Linear regressions of data with a pCO2 < 393? Please be more precise: give the name of the regression and the statistical parameters: e.g. Linear regression, R^2 = 0.90, P < 0.01, N = 10*

These are dynamic regression models described in our methods. The model parameters are stated in the text at L229-230.

22. *Table 1: please add two extra columns, one with the averaged alkalinity, and one withthe number of observations per lake (N).*

We have edited Table 1 as suggested.

23. *Fig. 1: x-axis label should be "Chl a (ug L-1)"*

Figure 1 (now Figure 3) axis label has been corrected.

**Reviewer 3.**

24. *A simple correlation of d13C values with Chl a concentration cannot be used in this study to predict CCM activity.*

This was not our intent. The purpose of this figure was only to visualize an increase in phytoplankton community d13C values with chlorophyll a concentration, which is commonly used as a proxy for phytoplankton biomass. We have edited this figure to only show the correlation between these two variables, rather than a linear regression.

25. *The authors describe the function of the CCM and how this could potentially change the isotopic signature of the cells (see line 91). Recent papers by Eichner et al 2015 and Raven and Beardal 2015 include internal cycling and loss terms of CO2. These two paper directly affect the interpretation of the data in this MS and should be introduced and discussed. Additional, a paper by Kranz et al 2015 showed the change in epsilon 13C during a bloom of diatoms. These authors also measured CCM parameters directly, seeing a switch from CO2 to HCO3- uptake at low CO2 conditions. However, this study used a model (Hopkinson) to predict the changes in d13C POC due to the switch to HCO3- uptake. The authors could contribute less than 0.5 permill change in the d13C signal to the switch in the inorganic carbon source. Together with the findings by Eichner et al 2015 and Raven and Beardall (2015). I feel that the authors have be aware that isotopic signal of organic matter are not necessarily driven by the uptake of different carbon species, but largely are affected by other cellular processes such as leakage as well as the external d13C DIC. Additionally, different species have different isotopic compositions – do the authors know if the lakes have similar phytoplankton communities?*

We have updated the Introduction to include a more extensive discussion of leakage and effects on isotopic composition (L80-86). We have included community composition data to demonstrate that these communities are dominated by cyanobacteria.

26. *In the method section the authors do not specifically mention how they obtained the biomass measured. Please be more precise in this and also mention how much of the organic material might have been detritus from other sources.*

The previous version of our manuscript used chlorophyll a as a proxy for phytoplankton biomass. We have updated the current manuscript to include biomass calculated from microscopic counts. Methods are detailed at L179-183. We manually removed zooplankton and detritus from filtered samples using a dissecting scope and are confident that the material measured was phytoplankton biomass (L177-178).

27. *The authors have to include the data of TA, DIC, d13C DIC, pH into Table 1 for the reader to understand the dataset and the correlations given.*

We have included these data in Table 1.

28. *The title of the MS is a little farfetched. Neither does the study proof that CCMs maintain biomass in the lakes not did the authors show actual CCM activity. Please revise.*

The title has been updated as suggested by Reviewers 1 and 2.

29. *Line 113: What are the criteria for which the lakes have been chosen? Wouldn't it be sufficient to just mention that 16 lakes were sampled and then briefly describe their properties?*

Lakes were chosen along an orthogonal gradient of interannual variability in cyanobacteria dominance and watershed permeability. This has been clarified in the text at L129-131.

30. *Line 143-145. I feel that this short paragraph should move behind line 160.*

The paragraph has been moved as requested, now L173-175.

31. *Line 171 and 172: describe what alpha a and alpha b means (Temperature-dependent fractionation factors between CO2 and HCO3 (a) as well as HCO3 ▯ and CO32▯ (b).*

This has been clarified at L198-199.

32. *Fig 1: Despite being significant, the predictive power of the dataset is relatively low! How would the dataset look like, if you use epsilon vs. Chl a. I feel that this would be more appropriate especially after reading how d13C seems to change in the different lakes.*

As mentioned above, the intent of this figure was not to show predictive power, and we agree that it may not have been appropriate to fit a regression line in this case. We have edited the figure to only show the correlation between these two variables.

33. *Line 220: Please rephrase: "This mechanism likely provides a competitive: : :" The authors refer to decreased fractionation as a mechanism, yet the fractionation calculated is the result of cellular mechanisms such as enhanced HCO3- uptake and/or enhanced CO2 leakage. Maybe rephrase to: "The cellular mechanisms which led to the decrease in fractionation under low pCO2 likely provide..."*

The text has been edited as suggested, now at 256-258.

[revised manuscript text omitted]

---

## Author Response (AR2)

Thank you again to all reviewers whose comments have substantially improved our manuscript. Final revisions are summarized below. Authors' responses to reviews are in italics.

**Reviewer 1**

I think the manuscript has improved a lot in clarity and focus after the revision. I am satisfied with the response to the issues I raised, I only have a few suggestions for technical corrections stated below.

1. Throughout the text, the word 'cyanobacteria' is sometimes written with a capital C and sometimes with a lower case c. Please be consistent.

   *Resolved*

2. Line 83: "HCO3-" misses sub- or superscripts.

   *Resolved*

3. Line 117: the use of the word 'monoculture' is incorrect here. A monoculture implies a single species. Please adapt this sentence.

   *Wording changed to: "...as community becomes dominated by phytoplankton using CCM."*

4. Line 171/174: please give the definition of VPDB the first time it is used in the text.

   *Resolved*

5. Figs 3-5: Could you please provide the equations of the regression curves in these figures?

   *We have included equations for figures 4 and 5 on the figures. We did not include a linear equation for Figure 3 because it is a correlation.*

6. Lines 233-234: The regression curve through the data in Fig. 5 looks like a parabolic curve that shows a negative relationship between the stable isotopic composition of the DIC pool and photosynthetic fractionation at high delta 13DIC. This must be one of the risks of using a nonlinear dynamic regression. Why not use a saturation function, or a linear regression?

   *We have updated this analysis, plot (Figure 5), and corresponding methods (L209) & results (L237-239) using a linear regression.*

7. Supporting Information: Could you please use the same range for each x-axis and y-axis (preferably consistent with the top panel of Fig. 4 of the main text)?

*We have edited the axes in the Supporting Information to be consistent with Fig. 4 in the main text.*

**Reviewer 2**

I thank the authors for revising this MS. The topic on isotope fractionation by freshwater algae is a complex one by itself and a multi-location study like the one presented combined with d13C discussion does not make it easy for the reader to follow. Although I'm not an expert on freshwater microalgae and fractionation processes I feel that some important aspects are missing in the discussion.

1) Lakes are like semi-closed systems and the photosynthetic fractionation affects the seawater d13C values and vice versa. The reason why fractionation is minimal when CO2 gets depleted (no source material left). The authors should elaborate on these processes.

*We have added a discussion of diffusive limitation at L309-317.*

2) Another general aspect should be addressed: In line 266-272 as well as in the following discussion the authors describe the effect of biotic processes on the source C (CO2/HCO3). How can one conclude on CCM fractionation processes and CO2 sources when the d13C values might be changing constantly due to the multitude of other processes affecting the system?

*We have updated the Discussion section to clarify these points. In this manuscript version and previous ones, we have attempted to constrain potential sources of DIC by documenting literature ranges of sources and processes (i.e., atmosphere, mineral dissolution, methanogenisis, microbial respiration of terrestrially-derived DOM) that could be responsible for our measured range of d13C values. We have clarified in the paragraph beginning at L287 that regardless of the source of DIC -- which cannot be definitively identified, only constrained – a heavy DIC pool combined with decreased photosynthetic fractionation points to active uptake of bicarbonate when $CO_2$ is depleted from the water column, particularly because in this pH range, geochemical processes dictate that the dominant inorganic carbon species should be bicarbonate, which cannot be taken up by passive diffusion.*

Other than these two general remarks/questions and a request to maybe simplify the paper (which seems hard) I see no issue for publication.

*In addition to the changes described above, we have revised and attempted to simplify*
*the Discussion by reorganizing the order of paragraphs to improve the flow. Specifically, we*
*moved the paragraph previously beginning at line 285 to (new) line 281 so that the discussion of*
*patterns in northern temperate lakes follows more logically from discussion of other studies.*

[revised manuscript text omitted]